# Longitudinal Analysis of Urinary Cytokines and Biomarkers in COVID-19 Patients with Subclinical Acute Kidney Injury

**DOI:** 10.3390/ijms232315419

**Published:** 2022-12-06

**Authors:** Gustavo Casas-Aparicio, Claudia Alvarado-de la Barrera, David Escamilla-Illescas, Isabel León-Rodríguez, Perla Mariana Del Río-Estrada, Mauricio González-Navarro, Natalia Calderón-Dávila, Rossana Olmedo-Ocampo, Manuel Castillejos-López, Liliana Figueroa-Hernández, Amy B. Peralta-Prado, Yara Luna-Villalobos, Elvira Piten-Isidro, Paola Fernández-Campos, Alejandro Juárez-Díaz, Karolina Piekarska, Santiago Ávila-Ríos

**Affiliations:** 1Centro de Investigación en Enfermedades Infecciosas, Instituto Nacional de Enfermedades Respiratorias Ismael Cosío Villegas, Calzada de Tlalpan 4502, Col. Sección XVI, Ciudad de Mexico 14080, Mexico; 2Dirección de Medicina, Fundación Clínica Médica Sur. Puente de Piedra 29, Col. Toriello Guerra, Ciudad de Mexico 14040, Mexico; 3Departamento de Enseñanza, Instituto Nacional de Enfermedades Respiratorias Ismael Cosío Villegas, Calzada de Tlalpan 4502, Col. Sección XVI, Ciudad de Mexico 14080, Mexico; 4Médica Santa Carmen, Periférico Sur 5580, Local B, Col. El Caracol, Ciudad de Mexico 04739, Mexico; 5Unidad de Epidemiología Hospitalaria e Infectología, Instituto Nacional de Enfermedades Respiratorias Ismael Cosío Villegas, Calzada de Tlalpan 4502, Col. Sección XVI, Ciudad de Mexico 14080, Mexico; 6Laboratorio Clínico, Instituto Nacional de Enfermedades Respiratorias Ismael Cosío Villegas, Calzada de Tlalpan 4502, Col. Sección XVI, Ciudad de Mexico 14080, Mexico

**Keywords:** acute kidney injury, COVID-19, cytokines, epidermal growth factor, neutrophil gelatinase-associated lipocalin, tissue inhibitor of metalloproteinases-2, insulin-like growth factor binding protein, kidney stress biomarkers

## Abstract

In hospitalized COVID-19 patients, disease progression leading to acute kidney injury (AKI) may be driven by immune dysregulation. We explored the role of urinary cytokines and their relationship with kidney stress biomarkers in COVID-19 patients before and after the development of AKI. Of 51 patients, 54.9% developed AKI. The principal component analysis indicated that in subclinical AKI, epidermal growth factor (EGF) and interferon (IFN)-α were associated with a lower risk of AKI, while interleukin-12 (IL-12) and macrophage inflammatory protein (MIP)-1β were associated with a higher risk of AKI. After the manifestation of AKI, EGF and IFN-α remained associated with a lower risk of AKI, while IL-1 receptor (IL-1R), granulocyte-colony stimulating factor (G-CSF), interferon-gamma-inducible protein 10 (IP-10) and IL-5 were associated with a higher risk of AKI. EGF had an inverse correlation with kidney stress biomarkers. Subclinical AKI was characterized by a significant up-regulation of kidney stress biomarkers and proinflammatory cytokines. The lack of EGF regenerative effects and IFN-α antiviral activity seemed crucial for renal disease progression. AKI involved a proinflammatory urinary cytokine storm.

## 1. Introduction

The kidney damage induced by SARS-CoV-2 is multifactorial; it can directly infect the kidney podocytes and proximal tubular cells, and based on an angiotensin-converting enzyme 2 (ACE2) pathway, it can lead to acute tubular necrosis, protein leakage in Bowman’s capsule, collapsing glomerulopathy and mitochondrial impairment [1], but viral-induced dysregulation of the immune responses can also cause kidney damage. In COVID-19 patients who require hospitalization, disease progression and severity may be driven by the over-activation of innate immune pathways, which results in the release of inflammatory cytokines and chemokines, and a corresponding depletion of several lymphocyte populations [2,3,4]. The plasma levels of inflammatory mediators, such as interleukin-2 (IL-2), IL-7, IL-10, interferon-gamma-inducible protein-10 (IP-10), granulocyte colony-stimulating factor, macrophage inflammatory protein-1α, tumor necrosis factor-α and monocyte chemoattractant protein 1, are significantly higher in patients with severe COVID-19 than in those with mild disease [5]. These virus-induced cytokines may exert indirect effects on renal tissue, such as hypoxia, shock and rhabdomyolysis [6].

Acute kidney injury (AKI) is a heterogeneous group of conditions characterized by a sudden decrease in glomerular filtration rate (GFR), manifested by an increase in serum creatinine (sCr) concentration or oliguria, and classified by stage and cause [7]. AKI occurs in 36–46% of hospitalized COVID-19 patients and in rates as high as 76% of those admitted to intensive care [8,9]. The diagnosis of AKI is based on a decrease in kidney function. Serum creatinine and urine output are the current gold standards for the evaluation of kidney function, but sCr is a low-sensitivity method because nearly 50% of GFR must be lost before an increase in sCr is detectable; and decreased urine output lacks specificity since it may be triggered by kidney hypoperfusion, without direct damage to the kidney [10]. Subclinical AKI involves the elevation of kidney biomarkers in the absence of changes in sCr and urine output [11]. The complex molecular changes and early tubular damage of subclinical AKI may only be identified by the aberrant expression of these structural kidney biomarkers. Therefore, understanding the fundamental molecular pathways and pathophysiology of this initial phase of kidney injury in COVID-19 is necessary for the development of diagnostic targets and effective therapies.

Considering that cytokines have been described in the plasma of patients with COVID-19-associated AKI, here we explored these molecules in the urine of patients with subclinical AKI who developed AKI on subsequent days. This way, we were able to describe the immunoregulatory proteins affecting the kidney before and after the development of AKI. Therefore, this longitudinal study was aimed at exploring the role of urinary cytokines and their relationship with the kidney stress biomarkers neutrophil gelatinase-associated lipocalin (NGAL), the tissue inhibitor of metalloproteinases-2 (TIMP-2) and the insulin-like growth factor binding protein 7 (IGFBP7) in a cohort of patients with severe COVID-19 and subclinical AKI.

## 2. Results

### 2.1. Characteristics of Study Participants

During the period between May and August 2020, a total of 420 individuals were admitted to critical areas of the National Institute of Respiratory Diseases (INER). Of those, 69 were negative for SARS-CoV-2 infection; in 44 SARS-CoV-2 infection could not be confirmed; 60 remained in the emergency room due to hospital saturation and 20 died there. Informed consent could not be obtained from 196 patients. We thus included 51 patients who provided informed consent for participating in the study (Figure 1). 

Of those, 30 were males (58.8%), the median age was 53 years (IQR: 40–61), 14 had systemic arterial hypertension (27.5%), 16 had diabetes (31.4%), and 21 were obese (41.2%). Of the 51 individuals studied, 28 had subclinical AKI (elevation of kidney biomarkers in the absence of changes in sCr and urine output) on admission to critical care areas (day 1) and developed AKI within the following 48 h (the AKI group, 54.9%), and 23 did not develop AKI (the non-AKI group, 45.1%). The AKI group had a higher frequency of hypertension (42% vs. 8.7% in the non-AKI group, *p* = 0.01), and had a higher SOFA score (4 in the AKI group (IQR: 3–7) vs. 3 in the non-AKI group (IQR: 2–6, *p* = 0.01).

Since day 1, the expression of kidney stress biomarkers was significantly higher in the AKI group, namely [TIMP2] × [IGFBP-7] 0.16 (ng/mL)^2^/1000 (IQR: 0.05–0.32) in the AKI group vs. 0.07 (ng/mL)^2^/1000 (IQR: 0.04–0.11) in the non-AKI group (*p* = 0.045); and NGAL 54.7 ng/mL (IQR: 36.4–117.4) in the AKI group vs. 32.4 (IQR: 14–40.05) in the non-AKI group (*p* = 0.002). In contrast, renal function had similar values in both groups on day 1 and became significantly lower in the AKI group on day 5, as indicated by sCr 0.79 mg/dL (IQR: 0.67–1.13) in the AKI group vs. 0.60 mg/dL (IQR: 0.51–0.75) in the non-AKI group (*p* = 0.01), and CKD-EPI 99 mL/min/1.73m^2^ (IQR: 69.5–109.5) in the AKI group vs. 110 mL/min/1.73m^2^ (IQR: 103–121) in the non-AKI group (*p* = 0.01, Table 1).

Concentrations of urinary cytokines were similar in both groups, except for the higher level of RANTES in the AKI group on day 1 (17.4 pg/mL, IQR: 13.2–25.4 vs. 10.7 pg/mL, IQR: 7.9–12.8 in the non-AKI group, *p* = 0.01, *q* = 0.14), and lower level of epidermal growth factor (EGF) in the AKI group since day 1 (2044 pg/mL, IQR: 719–4083 vs. 5568, IQR: 5056–5974 in the non-AKI group, *p* = 0.014, *q* = 0.04, Table 2). 

### 2.2. Performance of Biomarkers and Cytokines as Predictors of AKI 

Based on the highest area under the receiver-operating characteristics curve (AUC), specificity and accuracy values, the cytokines and biomarkers with the best performance for the prediction of AKI were EGF at a cutoff of 4600 pg/mL (AUC = 0.788, 95% confidence interval (CI): 0.597–0.979, *p* = 0.014) and NGAL at a cutoff of 40 ng/mL (AUC = 0.797, 95% CI: 0.668–0.927, Table 3).

### 2.3. Higher Levels of Urinary EGF Were Protective for AKI

The univariate analysis indicated that patients with AKI had a higher frequency of hypertension (odds ratio, OR = 11.42, 95% CI: 1.15–113, *p* = 0.037), and EGF > 4600 pg/mL was associated with a lower risk of AKI (OR = 0.05, 95% CI: 0.008–0.40; *p* = 0.004). After adjusting for possible confounding variables, only EGF > 4600 pg/mL remained associated with a lower risk of AKI (OR = 0.095, 95% CI: 0.01–0.81, *p* = 0.031, Table 4).

### 2.4. Correlation of Urinary EGF with Kidney Stress Biomarkers 

We explored the relation of urinary EGF and RANTES with kidney stress biomarkers on day 1. EGF had an inverse correlation with [TIMP2] × IGFBP7] (*R* = −0.73, *p* < 0.001) and with NGAL (*R* = −0.63, *p* = 0.002, Figure 2). RANTES only had a moderate correlation with NGAL (R = 0.37, *p* = 0.008).

### 2.5. Imputation of Cytokine and Chemokine Missing Values 

Cytokine and chemokine values outside the ranges specified by the Luminex manufacturer were treated as missing values in the data set, and these values were imputed. The numbers and proportions of missing values that were imputed for each cytokine are shown in Appendix A. 

### 2.6. Principal Component Analysis of Cytokines 

The principal component analysis (PCA) of cytokines indicated that the first five principal components (PC) had eigenvalues > 2 (PC 1: 6.48; PC 2: 4.82; PC 3: 3.88; PC 4: 3.27; PC 5: 2.08) and explained > 70% of the accumulated variance (73.43%, Appendix A). 

After adjusting for confounding variables, the PC1 (encompassing EGF and interferon (IFN)-α) was associated with lower odds of AKI (OR = 0.24, 95% CI: 0.07–0.78, *p* = 0.017) and the PC4 (encompassing IL-12 and macrophage inflammatory protein (MIP)-1β) was associated with higher odds of AKI (OR = 51.09, 95% CI: 2.12–1233, *p* = 0.015) on day 1. The PC1 was also associated with lower odds of AKI on day 5 (OR = 0.09, 95% CI: 0.01–0.74; *p* = 0.024) and the PC2 (encompassing IL-1 receptor (IL-1R), granulocyte-colony stimulating factor (G-CSF), IP-10 and IL-5) was associated with higher odds of AKI (OR = 7.7, 95% CI: 1.06–55.74, *p* = 0.043, Table 5). 

## 3. Discussion

The cytokine profile of patients with COVID-19-associated AKI has been studied in serum samples, but there is limited information derived from urinary samples reflecting what occurs in the kidney at the local level. Another particularity of our study is that we focused on subclinical AKI, while most studies have recruited patients with AKI stages 1 to 3. This way, we were able to describe the cytokine profiles before and after the development of AKI. On hospital admission, 54.9% of patients had subclinical signs of kidney dysfunction, which progressed to AKI within the following 48 h. This frequency was similar to that found in earlier studies, reporting 50% of AKI in COVID-19 at the intensive care unit [13]. In a previous study by our group, on the same cohort of participants [14], the frequency of AKI was 49% using the sCr criterion. Differences between studies may be explained by different AKI definitions, as here we used sCr and urine output, and it is well known that the use of both criteria increases AKI detection [15].

The samples obtained on day 1 of hospitalization describe the subclinical phase of AKI in those patients who would develop AKI on subsequent days. As indicated by logistic regression analysis, higher EGF levels were protective for AKI. The principal component analysis of cytokines in these samples indicated a lower risk of AKI with PC-1, encompassing IFN-α and EGF, and a higher risk of AKI with PC-4, encompassing IL-12 and MIP-1β. 

EGF promotes cellular proliferation, differentiation and survival [16]. In the kidney, EGF exerts several biological functions, such as regulation of cellular metabolism and glomerular hemodynamics, modulation of cell growth, and renal repair after injury [17]. Our results are in line with previous studies in COVID-19-hospitalized patients, indicating that elevated levels of urinary EGF are associated with a lower risk of AKI stage 3, new dialysis initiations or death [18]. In patients undergoing cardiac surgery, higher levels of urinary EGF were protective for CKD, and loss of EGF expression was found in those with AKI [19]. In patients with ischemic AKI, high urinary levels of EGF were associated with renal functional recovery, while low EGF levels were associated with AKI [20]. In chronic conditions requiring biopsy procedures, such as lupus nephritis [21] and type 2 diabetes [22], low urine EGF concentrations have been associated with histologic kidney damage and accelerated decline of renal function. Decreased EGF expression in COVID-19 patients with subclinical AKI might reflect low renal reserve and histological damage without these being clinically evident. However, various factors specific to COVID-19, including the use of mechanical ventilation, anticoagulation requirements, and logistical complexities, given the risk of viral transmission, make renal biopsies difficult to obtain in patients with suspected AKI [13]. 

We found that IFN-α was also associated with a lower risk of AKI on day 1. This finding is consistent with the notion that, in general, type I IFNs, such as IFN-α and IFN-β, are key antiviral factors. Host cells treated with IFN-α significantly inhibit SARS-CoV replication [23]. In fact, the ability of SARS-CoV-2 to downregulate the host IFN-I response is considered a viral strategy of immune evasion [24], and early expression of INF-α in patients with COVID-19 is associated with a lower risk of progression to severe forms [25]. Nevertheless, IFN-α is not only critical for protecting against viral infections, but it also promotes intracellular RNA degradation and virus clearance, induces tissue repair, and triggers a prolonged adaptive immune response [26,27,28]. Therefore, our results suggest that a higher expression of IFN-α, together with EGF had a protective role in the development of AKI. 

In samples obtained on day 1, we found that IL-12 and MIP-1β were associated with a higher risk of AKI. IL-12 is a potent, proinflammatory cytokine produced by antigen-presenting cells, typically in response to microbial pathogens. It is responsible for the induction and enhancement of cell-mediated immunity [29]. The relevant role of IL-12 in patients with severe COVID-19 has been described previously in a study showing that interleukin-10 and interleukin-12 levels, in combination with clinical variables, are key biomarkers associated with an increased risk of disease progression [30]. The MIP-1 chemokine plays a critical role in the recruitment of monocytes/macrophages during renal inflammation, and it is also important for the recruitment of T cells, macrophages, and dendritic cells during the development of chronic renal injury [31]. In summary, subclinical AKI was characterized by lower regenerative and antiviral protection conferred by EGF and IFN-α in the context of proinflammatory cytokines. 

It is worth mentioning that on day 1, the patients that would develop AKI on subsequent days already had a significant over expression of NGAL and [TIMP-2] × [IGFBP7], suggesting that the kidney was already under stress. At that point, kidney function was not significantly different in the group with subclinical AKI and the non-AKI group, indicating that sCr and urine output are not suitable for the diagnosis of subclinical AKI. EGF had an inverse relation with [TIMP2] × IGFBP7] and [NGAL], suggesting that a fragile tubule expressing low levels of EGF due to renal insult could represent an ideal setting for the up-regulation of kidney stress biomarkers. Our findings support previous reports describing elevated levels of [TIMP-2] × [IGFBP7] as predictors of adverse outcomes (e.g., death, dialysis, or progression to AKI in patients with septic shock) in various clinical conditions [32]; and imminent risk of AKI in critically ill patients [33]. In addition, our findings are consistent with a study reporting significantly higher NGAL levels in patients with COVID-19 without evidence of AKI on presentation who subsequently developed AKI stages 1 to 3 within seven days of admission, compared with those who did not develop AKI [34].

On day 5 of hospitalization, that is, when more than half of the patients had developed clinical AKI, we found that EGF and IFN-α were still associated with a lower risk of AKI, and IL-1R, G-CSF, IP-10 and IL-5 were associated with a higher risk of AKI. The IL-1R is primarily responsible for transmitting the inflammatory effects of IL-1; G-CSF stimulates the survival, proliferation, differentiation, and function of neutrophils and stimulates mobilization of hematopoietic stem cells to the bloodstream; IP-10 is a chemoattractant of monocytes, macrophages, T cells, NK cells and dendritic cells, and promotes T cell adhesion to endothelial cells; and IL-5 is a chemotactic and activating factor for B cells and eosinophils [35]. Thus, this frankly proinflammatory profile of urinary cytokines and chemokines, was no longer counterbalanced by the regenerative and antiviral effects of EGF and IFN-α. Our results are in accordance with the inflammatory cytokine and chemokine signatures described in the serum of hospitalized COVID-19 patients with end-organ injury, illustrating the potential deleterious effect of the cytokine storm on disease progression [2].

An important limitation of our study was the small sample size. Another study limitation was that patients with incomplete clinical files or those who were transferred to other hospitals due to the scarcity of intensive care beds were not included in the study, and this may represent a selection bias. In addition, our study was conducted at a national referral center for respiratory diseases receiving disproportionately more patients with severe COVID-19, and this represents a potential source of referral bias. The lack of pre-hospital baseline sCr measurements was also a study limitation because baseline sCr values were an estimation. The lack of sCr measurements prior to hospital admission often impairs the identification of underlying CKD and creates challenges for the reliable detection and staging of AKI. One study in which baseline sCr measurements were available, reported that 35% of patients with COVID-19-associated AKI had underlying CKD [36]. Another study limitation was that cytokine values outside the ranges specified by the Luminex manufacturer were treated as missing values, and the imputation of those values may represent a source of bias. Finally, an additional study limitation was that the origin of the cytokines and chemokines could not be established. As the kidney is an immunologically active organ, the observed markers could be found in the urine due to local renal production. Alternatively, the presence of the biomarkers in the urine may be a consequence of SARS-CoV-2 viremia. 

Taken together, our findings indicate that AKI was frequent in our cohort of hospitalized patients with COVID-19. Subclinical AKI was characterized by lower regenerative and antiviral protection conferred by EGF and IFN-α, in the context of proinflammatory cytokines. The lack of protective effects of EGF and IFN-α seemed to be crucial for the progression of renal disease. Up-regulation of NGAL and [TIMP-2] × [IGFBP7] was already significant, indicating that these kidney stress biomarkers may be useful for the diagnosis of subclinical AKI in patients with COVID-19. After the clinical manifestation of AKI, we observed an evident profile of inflammatory cytokines characteristic of the cytokine storm leading to disease progression, together with indicators of decreased renal function. Large randomized controlled clinical trials are required to further explore the role of kidney stress biomarkers and cytokines in critically ill patients with COVID-19.

## 4. Materials and Methods

### 4.1. Study Population

This prospective, longitudinal cohort study was conducted at the INER, the largest third-level institution designated by the Mexican Government for COVID-19. The Institutional Review Board approved the study (Approval No C26-20) and written informed consent was obtained from all participants. We included individuals that were admitted to intensive care with the diagnosis of severe pneumonia that was caused by SARS-CoV-2; who were 18 years of age or older; without AKI when a urine sample was collected; and with no history of chronic kidney disease (CKD) as indicated by interrogation of patients about CKD medical history and by an estimated eGFR greater than 60 mL/min/1.73m^2^ using the CKD-EPI equation [37]. Pregnant women were not included in the study. Patients with incomplete clinical records were excluded. SARS-CoV-2 severe pneumonia was defined by clinical data of respiratory distress, bilateral alveolar opacities in two or more lobes, a ratio of partial arterial oxygen pressure/inspired oxygen fraction (PaO_2_/FiO_2_) <300 mm Hg, and a positive result for SARS-CoV-2-real-time reverse transcription–polymerase chain reaction (rRT-PCR) assay in a nasopharyngeal swab [38]. The recorded variables included demographic and anthropometric variables, comorbidities, critical care variables, treatments, renal function indicators, blood chemistry, blood count, urinary kidney biomarkers and outcomes. We measured a panel of 28 cytokines in urine samples obtained on day 1 and day 5 of admission to critical care areas. 

### 4.2. Definition of Acute Kidney Injury 

The baseline sCr level was defined as the minimum inpatient value during the first 7 days of admission [39]. Diagnosis and staging of AKI were based on the Kidney Disease Improving Global Outcomes (KDIGO) criteria using serum creatinine (sCr) levels and urine output [12]. AKI stage 1 corresponded to an increase in sCr by ≥0.3 mg/dL within 48 h or an increase in sCr 1.5 to 1.9 times baseline within the prior 7 days; AKI stage 2 corresponded to an increase in sCr of 2.0–2.9 times baseline, and AKI stage 3 corresponded to an increase in sCr of ≥3 times baseline or the initiation of renal replacement therapy. 

### 4.3. Biomarker Determinations

The urine samples were collected on day 1 and on day 5. The urine was frozen at −80 °C within the first 30 min after sample collection. The urinary concentrations of TIMP-2 and IGFBP7 were determined using commercially available ELISA kits (Human TIMP-2 Quantikine ELISA Kit, R&D, Minneapolis, Minnesota, USA; Human IGFBP7 ELISA Kit, Abcam, Cambridge, UK) following manual instructions. The ELISA plates were read at O.D. of 450 and the calculations were done according to the signal that was given by the standard curve of each kit. NGAL determinations were done using the NGAL kit (Abbott, Chicago, IL, USA) according to the manual instructions and using the Abbott ™ ARCHITECT™ Analyzer (Chicago, IL, USA).

### 4.4. Cytokine Determinations 

The concentrations of cytokines and chemokines in urine were measured with a Luminex 200 instrument (Luminex Corporation, Austin, TX, USA) using xMAP (Multi-Analyte Profile) technology. Data were analyzed by Milliplex Analyst software to determine cytokine concentrations (VigeneTech, Carlisle, MA, USA). The Milliplex Human Cytokine Reagent Kit (Millipore, Billerica, MA, USA) was used to measure a cytokine panel consisting of fibroblast growth factor (FGF), G-CSF, granulocyte/macrophage-colony stimulating factor (GM-CSF), RANTES (regulated on activation, normal T cell expressed and secreted), MIP, monocyte chemoattractant protein-1 (MCP-1), EGF, hepatocyte growth factor (HGF), vascular endothelial growth factor (VEGF), IFN-α, IP-10, monokine induced by IFN-gamma (MIG), eotaxin; interleukin (IL)-1β, IL-1R, IL-10, IL-13, IL-6, IL-12, IL-17, IL-15, IL-5, IL-2, IL-7, IL-2R, IL-4 and IL-8.

### 4.5. Statistical Analysis

We performed descriptive statistics, including means and standard deviations for normally distributed continuous variables; medians and interquartile ranges (IQR) for non-parametric distributions and proportions for categorical variables. The comparisons of the group developing AKI during hospitalization vs. the group without AKI were made using a chi-squared test for categorical variables and Mann–Whitney U for continuous variables. The significantly different cytokines between groups were identified. The correlation of identified cytokines with kidney stress biomarkers was explored using the Spearman test. Given the large number of assessments for each group, we used the false discovery-based q-value statistic to correct for multiple hypothesis testing [40]. 

For each biomarker, the AUC with 95% CI was calculated, as well as the sensitivity, specificity, positive predictive value (PPV), negative predictive value (NPV), and the accuracy at three different cutoff values using urine samples that were collected on day 1. The cutoffs for each biomarker were selected based on the highest AUC, specificity, and accuracy for the prediction of AKI. Combinations of the top biomarkers were also explored. When combinations had no significant added value, the individual biomarkers were preferred. For all analyses, two-sided *p* values ≤ 0.05 were considered statistically significant.

Logistic regression analysis was used to identify the association between relevant covariates with AKI. We obtained age-stratified estimates considering 60 years and older as a vulnerable population. The variables were entered into the models when the alpha level of the risk factor was <0.20 in the univariate analysis. Age and gender were entered into the models regardless of the alpha level. All statistical tests were two-sided, and two-sided *p*-values of ≤0.05 were considered statistically significant. 

Values of cytokines outside the ranges specified by the Luminex manufacturer were treated as missing values in the data set. Missing values were imputed according to a practical guide for multiple imputation of missing data in nephrology [41], using PROC MI in SAS software, version 9.4 (Cary, NC, USA), the SAS system for Windows 10, and the mice package in R. 

Principal component analysis was performed to compress and simplify the size of the data set by keeping the most important information and analyzing the structure of the observations and the variables [42]. We retained components with eigenvalue >2 and a proportion of variance >70%. Stepwise regression was used to explore the association between principal components with AKI in urine samples collected on days 1 and 5. All statistical tests were two-sided and *p* values ≤ 0.05 were considered statistically significant. The PCA was conducted using R Studio 1.4.1717 (Boston, MA, USA).

## Figures and Tables

**Figure 1 ijms-23-15419-f001:**
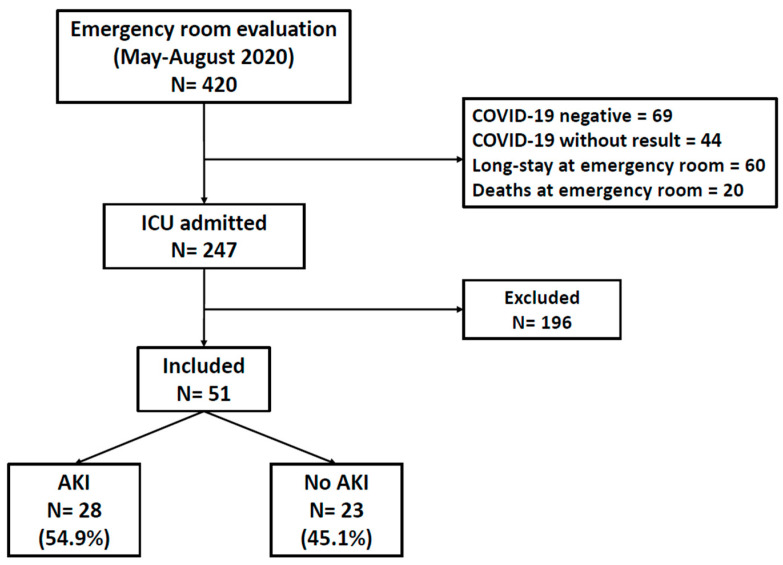
Study diagram. Numbers of individuals that were assessed for eligibility and individuals that were included in the study.

**Figure 2 ijms-23-15419-f002:**
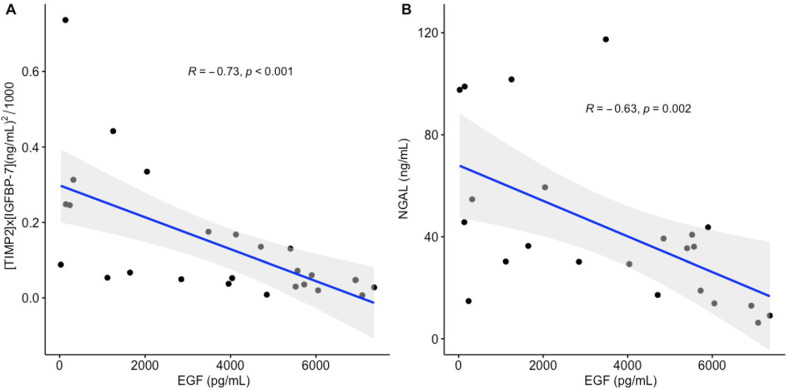
Correlation of urinary EGF with kidney stress biomarkers. (**A**) EGF had an inverse correlation with (tissue inhibitor of metalloproteinases-2, TIMP2) × (insulin-like growth factor binding protein, IGFBP7) (*R =* −0.73, *p* < 0.001); (**B**) EGF had an inverse correlation with neutrophil gelatinase-associated lipocalin, NGAL (*R* = −0.63, *p* = 0.002).

**Table 1 ijms-23-15419-t001:** Baseline characteristics of the study population.

Variable	Overall (*n* = 51)	AKI ** (*n* = 28)	Non-AKI (*n* = 23)	*p*
Age, years *	53 (40–61)	56 (40.5–61.5)	51 (40–55)	0.12
Male (*n* (%))	30 (58.8)	19 (68)	11 (47)	0.14
BMI, kg/m^2^ *	29.3 (25.9–31.6)	16 (57)	14 (46.7)	0.71
Comorbidities				
Obesity (*n* (%))	21 (41.2)	12 (43)	9 (39)	0.78
Diabetes (*n* (%))	16 (31.4)	8 (28)	8 (35)	0.63
Hypertension (*n* (%))	14 (27.5)	12 (42)	2 (8.7)	**0.01**
2 ≥ comorbidities (*n* (%))	21 (41.2)	14 (50)	7 (30.4)	0.15
Critical care variables				
IMV (*n* (%))	32 (62.7)	20 (71.4)	12 (52.2)	0.15
PaO_2_/FiO_2_ ratio, mmHg *	141 (108–187)	139 (101–162)	167 (132–220)	0.06
PEEP, cm H_2_O *	10 (9.5–14)	10 (10–14)	10 (10–12)	0.79
pH *	7.41 (7.33–7.44)	7.39 (7.34–7.44)	7.41 (7.33–7.46)	0.53
pCO_2_, mmHg *	38.0 (32.3–51.9)	42.1 (35–52.5)	37.5 (25–47)	0.15
SOFA score, points *	4 (2–6)	4 (3–7)	3 (2–6)	**0.01**
Treatments				
Vasoactive drugs (*n* (%))	16 (31.4)	10 (62.5)	6 (26)	0.46
Inotropic drug (*n* (%))	2 (3.9)	1 (50)	1 (4.3)	0.88
Systemic steroids (*n* (%))	25 (49)	13 (52)	12 (52)	0.68
Hydroxychloroquine (*n* (%))	7 (13.7)	5 (71.4)	2 (8.69)	0.34
Lopinavir/Ritonavir (*n* (%))	12 (23.5)	4 (33.3)	8 (34.7)	0.08
Nephrotoxic drugs (*n* (%))	1 (2)	0 (0.0)	1 (4.34)	0.26
Renal function indicators				
Serum creatinine, mg/dL, Day 1 *	0.60 (0.50–0.72)	0.61 (0.51–0.76)	0.60 (0.49–0.70)	0.58
eGFR, mL/min/1.73m^2^, Day 1 *	112.38 (98.45–121.40)	113 (96–118)	111 (106–124)	0.73
Serum creatinine, mg/dL, Day 5 *	0.69 (0.54–0.88)	0.79 (0.67–1.13)	0.60 (0.51–0.75)	**0.01**
eGFR, mL/min/1.73m^2^, Day 5 *	104 (90–117.50)	99 (69.5–109.5)	110 (103–121)	**0.01**
Final serum creatinine, mL/min *	0.67 (0.55–0.97)	0.74 (0.61–1.09)	0.61 (0.52–0.76)	0.13
Final eGFR, mL/min/1.73m^2^ *	104 (90–118-50)	102 (88.5–113)	107 (102–121)	0.07
Laboratories at Day 1				
Hemoglobin, g/dL *	13.3 (12.6–14.9)	13.4 (12.5–15)	13.3 (12.7–14.4)	0.99
Leucocytes, 10 ×^3^ mm^3^ *	8.9 (6.3–13.4)	9.3 (7.8–12.1)	7.8 (5.1–13.5)	0.22
Lymphocytes, 10 ×^3^ mm^3^ *	0.8 (0.6–1.0)	0.8 (0.45–1)	0.8 (0.65–1)	0.37
Platelets, 10 ×^3^ mm^3^ *	272 (219–329)	272 (216–362)	272 (219–314)	0.75
Lactate dehydrogenase, U/mL *	387 (299–557)	386 (314–544)	397 (265–572)	0.37
Total bilirubin, mg/dL *	0.47 (0.37–0.63)	0.56 (0.40–0.88)	0.43 (0.34–0.54)	0.07
Creatine phosphokinase, U/L *	140 (39–443)	224.5 (79.5–739.5)	60 (35.5–291)	**0.02**
D-dimer, µg/mL *	0.91 (0.42–2.50)	1.08 (0.55–4.27)	0.49 (0.34–1.29)	0.04
C-reactive protein, mg/dL *	16.5 (10–27.6)	18 (13–31.9)	13.2 (10–22.4)	0.05
Fibrinogen, mg/dL *	734 (580–821)	742 (604–805)	685 (565–786)	0.41
Procalcitonin, ng/mL *	0.39 (0.11–0.92)	0.62 (0.35–1.26)	0.14 (0.08–0.34)	**0.01**
Troponin, pg/mL *	5.6 (3.3–37)	11.8 (3.8–37)	3.9 (1.5–9)	**0.04**
Ferritin, ng/mL *	745 (358–1883)	710 (304–2491)	756 (457–953)	0.83
Urinary kidney biomarkers				
TIMP-2, ng/mL, Day 1 *	5.64 (3.03–9.02)	6.16 (3.49–9.66)	5.07 (3.01–7.24)	0.26
TIMP-2, ng/mL, Day 5 *	5.21 (3.31–7.81)	4.16 (3.18–7.36)	5.26 (3.65–12.03)	0.11
IGFBP7, ng/mL/1000, Day 1 *	13.83 (8.60–24.65)	17.43 (9.35–30.20)	12.50 (7.04–19.92)	0.12
IGFBP7, ng/mL/1000, Day 5 *	22.75 (12.63–46.20)	28.44 (13.05–77.78)	16.53 (11.93–32.35)	0.22
[TIMP2] × [IGFBP7], (ng/mL)^2^/1000, Day 1 *	0.085 (0.05–0.24)	0.16 (0.05–0.32)	0.07 (0.04–0.11	**0.04**
[TIMP2] × [IGFBP7], (ng/mL)^2^/1000, Day 5 *	0.14 (0.05–0.28)	0.13 (0.05–0.32)	0.14 (0.06–0.19)	0.99
NGAL, ng/mL, Day 1 *	39.3 (19.2–98.5)	54.70 (36.40–117.40)	32.40 (14–40.05)	**0.00**
NGAL, ng/mL, Day 5 *	33 (14–106.2)	35.20 (17.55–118.95)	20.50 (11.45–52.50)	0.07
Outcomes				
Day in hospital	16 (12–27)	20.5 (14.5–28)	13 (10.5–22.5)	**0.03**
Days on IMV	13 (10–22.2)	13 (12–22)	10.5 (8–26)	0.43
Mortality (*n* (%))	10 (19.6)	7 (25)	3 (13)	0.28

Acute kidney injury (AKI); body mass index (BMI); invasive mechanical ventilation (IMV); partial arterial oxygen pressure/inspired oxygen fraction (PaO_2_/FiO_2_); partial pressure of carbon dioxide (pCO_2_); positive end-expiratory pressure (PEEP); sequential organ failure assessment (SOFA); estimated glomerular filtration rate (eGFR); Systemic steroids: dexamethasone, methylprednisolone; prednisone; neutrophil gelatinase-associated lipocalin (NGAL); tissue inhibitor of metalloproteinases-2 (TIMP-2); insulin-like growth factor binding protein 7 (IGFBP7). * Data are expressed as medians (interquartile ranges). Comparisons of the AKI group vs. the non-AKI group were made using the chi-squared test for categorical variables and Mann–Whitney U for continuous variables. Bold values denote statistical significance at the *p* ≤ 0.05 level. ** Diagnosis and staging of AKI were based on the Kidney Disease Improving Global Outcomes (KDIGO) criteria using serum creatinine (sCr) levels and urine output [12].

**Table 2 ijms-23-15419-t002:** Urinary cytokines and chemokines.

Cytokine/ Chemokine (pg/mL)	Overall (*n* = 51)	AKI (*n* = 25)	No-AKI (*n* = 26)	*p* ^a^	*q* ^b^
FGF * Day 1	3.21 (1.78–9.14)	2.61 (1.74–7.31)	4.52 (1.97–11.81)	0.30	0.89
FGF * Day 5	2.86 (1.44–6.64)	2.43 (1.42–6.64)	3.75 (1.74–6.09)	0.76	0.97
IL-1β * Day 1	10.17 (5.54–20.11)	9.77 (4.92–15.94)	17.21 (7.71–25.31)	0.16	0.84
IL-1β * Day 5	7.38 (3.78–12.41)	6.48 (3.60–12.29)	9.07 (5.92–12.73)	0.22	0.97
G-CSF * Day 1	173.98 (123.48–197.43)	171.05 (119.76–224.19)	176.49 (166.81–188.64)	0.51	0.89
G-CSF * Day 5	159.05 (93.93–199.12)	181.37 (109.50–211.79)	157.53 (94.46–191.30)	0.56	0.97
IL-10 * Day 1	1.94 (1.70–4.48)	1.84 (1.70–3.16)	4.48 (1.88–13.35)	0.09	0.60
IL-10 * Day 5	3.52 (2.13–5.03)	3.52 (2.47–4.72)	2.59 (2.14–4)	0.73	0.97
IL-13 * Day 1	6.30 (3.01–9.28)	5.56 (2.67–7.87)	7.12 (4.03–9.93)	0.51	0.89
IL-13 * Day 5	5.68 (2.93–7.83)	5.33 (2.64–7.65)	6.05 (3.07–8.05)	0.65	0.97
IL-6 * Day 1	7.46 (3.57–15.43)	6.15 (3.71–14)	10.37 (3.39–19.61)	0.27	0.89
IL-6 * Day 5	6.16 (2.22–12.98)	5.92 (2.54–13.14)	6.97 (2.11–10.22)	0.90	0.99
IL-12 * Day 1	2.22 (1.73–3.40)	2.19 (1.75–3.26)	2.38 (1.73–3.27)	0.80	0.89
IL-12 * Day 5	2.26 (1.51–4.41)	1.83 (1.33–2.37)	2.80 (1.74–5.65)	**0.01**	0.41
RANTES * Day 1	13.83 (9.38–22.67)	17.40 (13.2–25.4)	10.7 (7.9–12.8)	**0.01**	0.14
RANTES * Day 5	15.98 (12.93–25.14)	15.62 (11.14–28.60)	16.11 (12.94–22.84)	0.82	0.99
Eotaxin * Day 1	2.20 (1.12–4.24)	2.52 (1.17–4.24)	1.87 (1.13–4.62)	0.82	0.89
Eotaxin * Day 5	3.50 (1.24–6.29)	3.26 (1.84–6.18)	4.10 (0.68–6.24)	0.64	0.97
IL-17A * Day 1	3.05 (2.34–5.23)	3.19 (2.21–8.20)	2.90 (2.53–3.19)	0.37	0.89
IL-17A * Day 5	3.04 (2.50–3.21)	3.08 (2.60–3.63)	2.99 (2.44–3.20)	0.60	0.97
MIP-1α * Day 1	21.35 (10.97–35.89)	21.35 (9.61–30.61)	20.85 (11.79–35.89)	0.69	0.89
MIP-1α * Day 5	11.79 (6.04–29.27)	9.08 (4.90–29.27)	11.79 (9.61–20.85)	0.52	0.97
GM-CSF * Day 1	0.65 (0.38–1.27)	0.75 (0.43–1.27)	0.59 (0.34–1.22)	0.46	0.89
GM-CSF * Day 5	0.68 (0.43–1.32)	0.53 (0.38–1.38)	0.74 (0.53.1.26)	0.44	0.97
MIP-1β * Day 1	3.27 (1.84–8.16)	3.27 (1.84–6.64)	3.25 (1.84–8.17)	0.79	0.89
MIP-1β * Day 5	4.32 (1.66–6.62)	2.87 (1.37–6.64)	4.37 (2.13–6.03)	0.65	0.97
MCP-1 * Day 1	137.22 (50.57–311)	148.43 (61.49–323.19)	121.02 (49.56–237.50)	0.62	0.89
MCP-1 * Day 5	124.6 (50.77–242.04)	116.29 (91.46–254.88)	143.23 (45.57–219.43)	0.88	0.99
IL-15 * Day 1	127.72 (82.62–196.61)	108.47 (79.62–240.22)	130.74 (93.11–193.87)	0.73	0.89
IL-15 * Day 5	140.19 (97.14–176.12)	145.27 (128.26–190.80)	103.37 (56.60–142.85)	0.12	0.97
EGF * Day 1	4083.27 (1218.24–5768.37)	2044 (719–4083)	5568 (5056–5974)	**0.01**	**0.14**
EGF * Day 5	4127.73 (1722.80–5494.66)	3934.24 (1552.42–5254.31)	4614.17 (3537.42–5424.23)	0.55	0.97
IL-5 * Day 1	0.77 (0.23–2.38)	0.91 (0.23–4.73)	0.67 (0.29–1.25)	0.34	0.89
IL-5 * Day 5	2.38 (0.46–5.85)	2.38 (0.35–6.40)	2.38 (1.44–2.71)	0.75	0.97
HGF * Day 1	41.18 (21.27–80.86)	40.12 (21.27–80.22)	41.18 (21.99–81.51)	0.80	0.89
HGF * Day 5	53.12 (28.68–79.97)	54.82 (28.84–70.20)	52.50 (32.45–79.97)	0.86	0.99
VEGF * Day 1	2.59 (0.82–4.41)	2.26 (0.82–3.79)	3 (0.70–4.74)	0.77	0.89
VEGF * Day 5	2.74 (0.77–4.41)	2.88 (0.95–3.76)	1.36 (0.77–4.60)	0.60	0.97
IFN-α * Day 1	10.90 (1.89–17.50)	10.90 (1.55–16.77)	13.64 (2.52–17.50)	0.27	0.89
IFN-α * Day 5	12.09 (1.92–17.21)	10.90 (1.92–14.39)	13.45 (2.08–18.82)	0.41	0.97
IL-1RA * Day 1	2829.72 (1209.25–5344.40)	2689.10 (1238.42–5813.46)	3421.78 (1263.64–4084.19)	0.90	0.93
IL-1RA * Day 5	1879.33 (1000.58–4591.50)	1675.03 (1071.18–3571.60)	2088.81 (830.36–4601.67)	0.72	0.97
IL-2 * Day 1	1.57 (1.27–2.51)	1.58 (1.45–2.27)	1.53 (1.12–2.23)	0.45	0.89
IL-2 * Day 5	1.83 (1.26–2.52)	1.79 (1.34–3)	1.88 (1.26–2.27)	0.98	1
IL-7 * Day 1	14.31 (4.35–29.41)	15.69 (4.65–30.05)	12.93 (4.54–27.03)	0.66	0.89
IL-7 * Day 5	10.41 (3.71–30.71)	12.59 (3.95–43.47)	6.78 (4.06–16.37)	0.42	0.97
IP-10 * Day 1	2.38 (2.04–6.06)	2.52 (2.02–7.33)	2.24 (2.19–3.63)	0.64	0.89
IP-10 * Day 5	2.93 (1.44–5.02)	3.06 (1.81–6.77)	2.78 (1.75–3.75)	0.49	0.97
IL-2R * Day 1	653 (197.59–1633.81)	649.43 (203.20–1429.30)	786.61 (193.19–1686.19)	0.84	0.89
IL-2R * Day 5	296.03 (116.97–990.57)	304.80 (121.13–1103)	296.03 (107.33–818.01)	0.89	1
MIG * Day 1	43.28 (21.73–52.93	39.96 (15.74–56.63)	46.12 (25.48–51.75)	0.67	0.89
MIG * Day 5	39.97 (21.73–61.45)	34.17 (21.73–60.93)	49.22 (22.28–61.45)	0.53	0.97
IL-4 * Day 1	70.19 (3.15)	6.36 (6.36–6.36)	6.36 (6.36–6.36)		
IL-4 * Day 5	10.29 (6.18)	34.98 (6.18–63.78)		1	1
IL-8 * Day 1	8.52 (2.66–28.19)	9.04 (2.63–28.60)	7.83 (3.47–26.93)	1	0.89
IL-8 * Day 5	15.09 (5.14–76.49)	10.81 (5.22–73.64)	16.42 (3.18–135.60)	0.61	0.97

Acute kidney injury (AKI); body mass index (BMI); systemic steroids: dexamethasone, methylprednisolone; prednisone; fibroblast growth factor (FGF); granulocyte colony-stimulating factor (G-CSF); granulocyte/macrophage-colony stimulating factor (GM-CSF); RANTES (regulated on activation, normal T cell expressed and secreted); macrophage inflammatory protein (MIP); monocyte chemoattractant protein-1 (MCP-1); epidermal growth factor (EGF); hepatocyte growth factor (HGF); vascular endothelial growth factor (VEGF); interferon alpha (IFN-α); interferon-gamma-inducible protein 10 (IP-10); monokine induced by IFN-gamma (MIG); interleukin (IL); IL-1 receptor (IL-1R). * Data are expressed as medians (interquartile ranges). ^a^ Mann–Whitney U test (*p* values ≤ 0.05 in bold). ^b^ Storey’s *q* for multiple comparisons (*q* values < 0.2 in bold).

**Table 3 ijms-23-15419-t003:** Performance of urinary biomarkers for prediction of acute kidney injury in COVID-19 patients.

Biomarker	AUC	95% CI	*p*	Cutoff	Sensitivity (%)	Specificity (%)	PPV (%)	NPV (%)	Accuracy (%)
Prediction of AKI on day 1									
EGF (pg/mL)	0.788	0.597–0.979	0.014	4600	80.00	81.82	74.58	85.99	81.09
[TIMP2] × [IGFBP7] (ng/mL)^2^/1000	0.672	0.521–0.823	0.036	0.2	42.86	95.65	86.79	71.52	74.53
NGAL (ng/mL)	0.797	0.668–0.927	0.014	40.0	68.00	73.91	63.47	77.60	71.55

Acute kidney injury (AKI); area under the receiver-operating characteristics curve (AUC); confidence interval (CI); positive predictive value (PPV); negative predictive value (NPV); neutrophil gelatinase-associated lipocalin (NGAL); tissue inhibitor of metalloproteinases-2 (TIMP-2); insulin-like growth factor binding protein 7 (IGFBP7).

**Table 4 ijms-23-15419-t004:** Risk factors for acute kidney injury in critically ill COVID-19 patients.

Variables	Unadjusted OR (CI 95%)	*p*	Adjusted OR (CI 95%)	*p*
Age > 60 years	2.29 (0.44–11.91)	0.324	0.84 (0.06–11.22)	0.901
Men	2.25 (0.34–14.61)	0.396	2.75 (0.28–26.68)	0.381
Hypertension	11.42 (1.15–113)	**0.037**	5.75 (0.32–103)	0.235
D-Dimer μg/ml	1.35 (0.18–10.0)	0.769	-	-
NGAL > 40 ng/ml	2.66 (0.46–15.25)	0.270	-	-
EGF > 4600 pg/mL *	0.05 (0.008–0.40)	**0.004**	0.095 (0.01–0.81)	**0.031**

Odds ratio (OR); confidence interval (CI); neutrophil gelatinase-associated lipocalin (NGAL); epidermal growth factor (EGF). Variables were entered into de model when the alpha level of the risk factor was less than 0.20. Age and gender were added into the model regardless of the alpha level. * Analysis of EGF using measurable variables. Bold values denote statistical significance at the *p* ≤ 0.05 level.

**Table 5 ijms-23-15419-t005:** Risk of acute kidney injury associated with principal components in critically ill COVID-19 patients.

Principal Component	Unadjusted OR (95% CI)	*p*	Adjusted OR (95% CI) *	*p*
Day 1				
PC-1: IFN-α; EGF	0.5 (0.23–1.11)	0.08	0.24 (0.07–0.78)	**0.01**
PC-2: IL-1R; G-CSF; IP-10; IL-5	7.14 (0.58–87.78)	0.12	15.95 (0.31–817)	0.16
PC-3: IL-10	0.45 (0.13–1.51)	0.19	-	-
PC-4: IL-12; MIP-1β	7.38 (0.95–57.18)	0.05	51.09 (2.12–1233)	**0.01**
PC-5: HGF; MCP-1; IL-6	0.39 (0.12–1.22)	0.10	-	-
Day 5				
PC-1: IFN-α; EGF	0.37 (0.14–0.97)	**0.04**	0.09 (0.01–0.74)	**0.02**
PC-2: IL-1R; G-CSF; IP-10; IL-5	1.81 (0.64–5.13)	0.26	7.7 (1.06–55.74)	**0.04**
PC-3: IL-10	6.46 (0.72–58.29)	0.09	-	-
PC-4: IL-12; MIP-1β	0.42 (0.11–1.57)	0.19	0.2 (0.02–1.73)	0.14
PC-5: HGF; MCP-1; IL-6	1.13 (0.41–3.12)	0.81	9.37 (0.97–90.1)	0.05

Principal component (PC); odds ratio (OR); confidence interval (CI); interferon alpha (IFN-α); epidermal growth factor (EGF); interleukin (IL); IL-1 receptor (IL-1R); granulocyte colony-stimulating factor (G-CSF); interferon-gamma-inducible protein 10 (IP-10); macrophage inflammatory protein-1 beta (MIP-1β); hepatocyte growth factor (HGF). Bold values denote statistical significance at the *p* ≤ 0.05 level. * Adjusted for age and sex.

## Data Availability

All data generated and analyzed during this study were included in a Appendix A.

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
