# Peer review of "Longitudinal Analysis of Urinary Cytokines and Biomarkers in COVID-19 Patients with Subclinical Acute Kidney Injury"

_ijms, 2022, doi:10.3390/ijms232315419_

Round 1

Reviewer 1 Report

Authors explored the role of urinary cytokines and their relationship with the kidney stress biomarkers: neutrophil gelatinase-associated lipocalin, tissue inhibitor of metalloproteinases-2 and insulin-like growth factor binding protein in COVID-19 patients without acute kidney injury (AKI) at study entry as well as their prognostic value after AKI development. The study is dominantly based on author’s results previously published in Biomolecules 2022, https://doi.org/10.3390/biom12020275 using the same cohort of patients and their baseline characteristics. They found that subclinical AKI was characterized by lower regenerative and antiviral protection conferred by EGF and IFN-α in a context of proinflammatory cytokines, and that lack of EGF and IFN-α protective effects was crucial for progression of renal disease. Although authors present new insight into AKI prognosis in COVID-19 patients, there is no crucial breakthrough in understanding the molecular mechanisms of AKI deterioration, while the conclusion is nearly to remarks previously published in Biomolecules. Authors should omit presentation of data published in the previous article. Therefore, I find the article with low priority for publishing in IJMS.

Reviewer 2 Report

Dear authors you will find attached my comments.

Round 2

Reviewer 1 Report

Authors sufficiently improved the manuscript. 

Reviewer 2 Report

I think you have made substantial changes.
It could be improved however
I think it can be published after minor changes possibly based on the comments
of the editors